# Clinical Efficacies of FLT3 Inhibitors in Patients with Acute Myeloid Leukemia

**DOI:** 10.3390/ijms232012708

**Published:** 2022-10-21

**Authors:** Moo-Kon Song, Byeong-Bae Park, Ji-Eun Uhm

**Affiliations:** 1Department of Hematology-Oncology, Hanyang University Hanmaeum Changwon Hospital, Changwon 51497, Korea; 2Division of Hematology-Oncology, Department of Internal Medicine, Hanyang University College of Medicine, Hanyang University Seoul Hospital, Seoul 04763, Korea

**Keywords:** FMS-like tyrosine kinase 3, acute myeloid leukemia, midostaurin, gilteritinib

## Abstract

FLT3 mutations are the most common genomic alteration detected in acute myeloid leukemia (AML) with a worse clinical prognosis. The highly frequent FLT3 mutations, together with the side effects associated with clinical prognosis, make FLT3 promising treatment targets and have provoked the advancement of FLT3 inhibitors. Recently, numerous FLT3 inhibitors were actively developed, and thus the outcomes of this aggressive subtype of AML were significantly improved. Recently, midostaurin and gilteritinib were approved as frontline treatment of AML and as therapeutic agents in the recurred disease by the United States Food and Drug Administration. Recently, numerous promising clinical trials attempted to seek appropriate management in frontline settings, in relapsed/refractory disease, or after stem cell transplantation in AML. This review follows numerous clinical trials about the usefulness of FLT3 inhibitors as frontline therapy, as relapsed/refractory conditioning, and as maintenance therapy of stem cell transplantation. The cumulative data of FLT3 inhibitors would be important clinical evidence for further management with FLT3 inhibitors in AML patients with FLT3 mutations.

## 1. Introduction

The numerous molecular conformations of acute myeloid leukemia (AML) have become obvious over the past decade. FMS-like tyrosine kinase 3 (FLT3) receptor-Internal tandem duplications (ITD) and tyrosine kinase (TK) mutations can rise and predict worse clinical outcomes [1].

FLT3 is a type III receptor tyrosine kinase (RTK) including KIT, FMS, and PDGFR. It consists of five immunoglobulin-like domains in the extracellular region, a juxtamembrane (JM) domain, a TK domain divided by a kinase insert domain, and a C-terminal domain in the intracellular region [2].

FLT3 is represented on lineage-restricted progenitor cells and thus FLT3 ligand (FL) could activate FLT3. Thus, induced FLT3 activation also stimulates the activations of numerous signal pathways [3]. In this pathway, FLT3 could be expressed on leukemic blasts, and activation of FL signaling could stimulate proliferation and reduce apoptosis of the leukemic blasts [2,4]. Previously, an ITD has been found in the JM domain-coding sequence of FLT3-ITD [5], while a missense point mutation at the D835 residue and deletions, and insertions in D835 were identified in **FLT3-TKD mutations** [6].

FLT3-ITD and -TKD mutations possess about 25 and 5% of AML patients [7,8,9]. Due to the fact that FLT3 mutations are the most common mutations with a worse outcome, FLT3 mutation is one among several significant molecular markers for management of AML [10]. Despite this disparity, until now standard 7 + 3 induction therapy has remained the front-line standard chemotherapy for AML.

Recently, the advent of FLT3 inhibitors has made possible the expansion of therapeutic schemes for application in FLT3-mutated AML patients. In 2017, the Food and Drug Administration (FDA) approved the addition of the targeted therapeutic agent midostaurin for patients treated with chemotherapy and for patients with newly diagnosed FLT3-mutated. Additionally, gilteritinib, a small-molecular inhibitor of FLT3, was approved by the FDA for the management of relapsed or refractory (R/R) AML with FLT3 mutation according to the randomized phase III ADMIRAL study.

Although FLT3 inhibitors have the potential to improve the clinical outcome of patients in a high-risk subset with FLT3 mutations, questions surrounding the therapeutic applications of these agents in several complicated conditions remain. FLT3 inhibitors were applicated for leading to an increment of therapeutic efficacies in FLT3 mutated AML patients. In this review, we would summarize the characteristics of FLT3 mutations and discuss frontline therapy, R/R management, and maintenance therapy involving FLT3 inhibitors.

## 2. FLT3 Genetic Aberrations

FLT3 is represented in the acute leukemic cells and the incidence of the mutation occurs in about 30% of patients [11]. The incidence of the FLT3-ITD mutation is about 25%, while the FLT3-TKD mutation is about 5%. Both FLT3 mutations are active mutations, associated with ligand-independent, or linked with FLT3 receptor signaling, thereby stimulating proliferation and survival of the leukemic cells. Signal aberration onsets in correlation with retained FLT3-ITD in the endoplasmic reticulum (ER), with trafficking of the receptor out of the ER-Golgi impaired by the presence of the duplicated domain [12].

Between mutation mechanisms, both FLT3 mutations read to systemic activation of FLT3 signaling, and the signal pathways between individual mutations were different. The presence of ITD could read to activation of TKD, and FLT3 signal by ITDs is aberrant, particularly stimulating STAT5 and the downstream effectors, such as Pim-1 [13].

Activation of FLT3-ITD mutation is relevant to STAT5, PI3K/Akt, and MAPK/ERK [13]. Meanwhile, activation of FLT3-TKD mutation is associated with Akt and ERK, not STAT5. Thus, FLT3-mutated AML clinically showed worse relapse-free survival (RFS) and overall survival (OS) [12].

Interestingly, prognostic scoring is affected by both the mutation of allele burden and co-existing other mutations. According to the FLT3 mutation risk stratification system, a high allelic ratio (high AR; high FLT3-ITD), typically identified as an FLT3-ITD to FLT3-WT ratio of ≥0.5, is correlated with poorer prognostic risk [14,15]. A low AR (low FLT3-ITD) is correlated with favorable risk in the intermediate-risk group with coincident nucleophosmin 1 (NPM1) mutation [10]. The associations are reflected in the 2017 European LeukemiaNet (ELN) risk stratification of AML [16]. Meanwhile, FLT3-TKD mutations mean point mutations within the receptor’s activation loop stabilizing and activating the kinase conformation. However, the clinical prognosis of FLT3-TKD mutation in AML risk assessment is less obvious.

## 3. Clinical Influence of FLT3 Aberrations in Newly Diagnosed or Relapsed/Refractory Settings

The FLT3-ITD mutated group is relevant to a notably poor outcome, with an increasing risk of relapse and decreasing OS. The analysis showed that FLT3-ITD mutated group had shorter OS and RFS. In numerous data analyses, only a weak association between FLT3-TKD and survival (OS or RFS) was shown [17,18,19].

The AR, ITD length, chromosome, and NPM1 mutation seem likely to further affect the clinical influence of FLT3-ITD mutations [20]. In some data, allelic burden reflected by the FLT3-ITD mutation might be associated with clinical outcomes. For instance, in a study evaluating the clinical impact of FLT3-ITD in subgroups with newly diagnosed AML with AR > 0.78 was significantly linked to lower OS and disease-free survival (DFS) [21]. In another similar study in a group with newly diagnosed AML with FLT3-ITD mutation, a high AR (>0.51) and FLT3-ITD insertion site in TKD1 were associated with low complete remission (CR) and worse prognosis [22]. Particularly, patients did not receive FLT3 inhibitors. In the RAFITY study, OS was increased by midostaurin therapy versus placebo in groups with high FLT3-ITD and low ITD allelic burdens, demonstrating that both groups with high and low mutant-to-wild type ratio might gain a profit from the addition of an FLT3 inhibitor [23]. However, other studies demonstrated that the risk of relapse is not correlated with AR [24]. In contrast, the FLT3-ITD burden was shown to reflect worse OS and RFS.

Clonal evolution such as FLT3-ITD is especially important because FLT3 mutations in relapsed conditions have been associated with inferior OS than that of FLT3-wild-type (WT). In particular, FLT3-ITD mutations at relapse were shown to be a negative prognostic factor independent of other factors in AML patients who failed induction therapy. Moreover, other data demonstrated that FLT3-ITD mutated AML patients showed a high relapse rate in salvage chemotherapy and even after a potentially curative allogeneic stem cell transplantation (allo-SCT).

The FLT3-ITD mutations in R/R AML is correlated with decreased durability of CR, increased risk of relapse, and lowered OS after standard-of-care (SOC) therapy [25,26,27].

## 4. FLT3 Inhibitor Classifications

The development of first-generation FLT3 inhibitors was not naturally associated with FLT3 specificity. Therefore, first-generation FLT3 inhibitors were relatively non-specific inhibitors with additional receptor targets such as c-Kit, platelet-derived growth factor receptor (PDGFR), and vascular endothelial growth factor receptor (VEGFR). The second-generation FLT3 inhibitors were developed to have more potent FLT3 inhibitory activity.

Moreover, FLT3 inhibitors are separated into type I and II classifications on the basis of the interaction mechanism with the receptor. All FLT3 inhibitors interrupt the binding of ATP to the tyrosine kinase domain, inhibiting phosphorylation and activation of the FLT3 receptor [6,28]. Type I inhibitors such as lestaurtinib, midostaurin, crenolanib, and gilteritinib, bind to FLT3 with active or inactive forms of molecular structures, while type II inhibitors such as sorafenib, quizartinib, and ponatinib, connect with a hydrophobic area of FLT3 in the only inactive form of molecular structure. In the genetic data, D835 aberrations in the TKD induce FLT3 to favor the active conformation, inhibiting the activity of type II FLT3 inhibitors to interact and block FLT3-TKD. Thus, type II FLT3 inhibitor therapy over time might select for D835 mutations and result in acquired resistance, while type I inhibitors could retain activity at both FLT3 mutations [29].

## 5. Front-Line Therapy for FLT3-ITD-Mutated AML

### 5.1. Sorafenib

Sorafenib is a first-generation inhibitor of FLT3 with various kinase inhibitory activities, such as RAF-1, VEGFR, c-KIT, PDGFR, ERK, and FLT3 [30,31]. Moreover, it has a potent anti-cancerous effect on AML cells with FLT3 mutation. The completed FLT3-ITD inhibited activity is with an IC50 of 69.3 ng/mL [32].

Numerous mechanisms can interpret the clinical impact of sorafenib in managing AML. First, sorafenib could stimulate secretion of IL-15 by AML cells with FLT-ITD mutations and increase the survival duration of AML patients with FLT3-ITD mutation [33]. Moreover, persistently low blast rate, highly proportioned CD8 + lymphocytes in the bone marrow (BM), and highly expressed COL4A3, TLR9, FGF1, and IL-12 genes have been shown in patients who received sorafenib therapy [34].

In addition, sorafenib was shown to inhibit Src kinase-mediated STAT3 phosphorylation and reduce the representation of apoptosis regulatory proteins, Mcl-1 and Bcl-2 [35]. Finally, sorafenib could decrease treatment-triggering of Smac mimetic-induced necroptosis in AML cells resistant to apoptosis [36].

Sorafenib in combination with conventional therapy was studied for AML therapy. Rolling et al. showed that a randomized, placebo-controlled SORAML trial investigates the addition of sorafenib to standard chemotherapy in 276 newly diagnosed AML patients younger than 60 years [37]. The addition of sorafenib led to improved event-free survival (EFS) (median, 21 vs. 9 months) compared to the placebo. Additionally, 3-yr EFS was superior to sorafenib (40% vs. 22%, *p* = 0.013). Moreover, RFS was higher in the sorafenib group (56% vs. 36%). However, OS was not different (Table 1). OS seemed to improve with sorafenib treatment in those with higher FLT3-ITD ARs or receiving transplants in the first CR, but the study was not powered to detect statistical significance.

Serve et al. showed a phase III study of sequential sorafenib therapy after intensive chemotherapy to older patients [38]. The data showed no differences between the overall response rate (ORR), EFS, and OS in the overall group and FLT3-ITD-positive group. Sorafenib with conventional chemotherapy was not beneficial in the elderly population.

AML cells with FLT3-ITD mutations showed meaningful responses in sorafenib therapy in several in vitro and ex vivo data. A phase I study of 50 patients with R/R AML, including 28 with FLT3-ITD mutations and 6 with both ITD mutated and TKD mutated, administered sorafenib therapy. Five of the patients with FLT3-ITD mutations achieved CR/CRi (complete responses [CR]/complete responses with incomplete hematologic recovery), while BM or peripheral blood (PB) blasts were significantly reduced in the other 17 patients [39]. Therefore, sorafenib showed significant activity in AML with FLT3-ITD mutations. However, to obtain more effective data, further data such as combination therapy with other agents are needed.

Sorafenib was combined with standard therapy in the SORAML phase II trial. In the study, 134 patients were randomized to be treated with sorafenib plus chemotherapy and the other 133 patients did not receive sorafenib [40]. In the data, 23 patients have AML with FLT3-ITD mutations (17%). Sorafenib group had higher 5-year EFS (41% vs. 27%; *p* = 0.011) and higher RFS (53% vs 36%; *p*= 0.035) compared to the placebo group. However, 5-year OS was not different (61 vs 53%; *p* = 0.282) (Table 1).

Furthermore, sorafenib has been studied combined with hypomethylating agents in FLT3-mutated patients. The phase II trial of sorafenib with 5-azacitidine (5-AZA) have been studied in FLT3-mutated AML patients with both R/R and newly diagnosed disease [41,42]. These data demonstrated hopeful efficacy and safety profiles but phase III trials are needed to further confirm the benefits.

### 5.2. Midostaurin

Midostaurin is a first-generation inhibitor targeting both ITD and TKD mutations with widely inhibition effect against protein kinase C, c-KIT, VEGFR, and PDGFR-β [43,44].

It is an orally administered type I inhibitor with broad specificities that include WT and mutated FLT3, VEGFR, c-KIT, and PDGFR. Midostaurin could prevent the signaling pathways regulated by several kinases, and thus growth arrest was led. It could be activated on FLT3-ITD and TKD-derived kinases. It is also metabolized by CYP3A4, leading to several drug interactions [45].

Midostaurin was introduced as a therapeutic agent for R/R AML patients and cytoreduced or killed PB leukemic blasts in FLT3-mutated or FLT3-WT patients. It was approved in 2017 by the US FDA for the management of AML patients with FLT3 mutations and has also recently been accepted for those newly diagnosed with the FLT3 mutation and or systemic mastocytosis.

In one data point, therapeutic schedules of different dosing midostaurin combined with chemotherapy in 69 AML patients younger than 60 years with FLT3 mutation or FLT-WT showed CR rates, 80% in a twice daily 50 mg dose cohort (*n* = 40) [43]. The 1-year OS possibility and 2-year OS possibility were 85% and 62% in the group with FLT3 mutation, and 78% and 52% in AML those with WT, respectively. The median OS of the FLT3 mutation group was not different from the WT group, resulting in the assumption that the added midostaurin counteracts the negative influence of the mutations and the rising clinical outcomes of the disease group.

The RATIFY study showed adding midostaurin to standard chemotherapy followed by maintenance therapy for 1 year led to a significant improvement in OS (74.7 months vs. 25.6 months, *p* = 0.009) and EFS (8.2 months vs 3.0 months, *p* = 0.002) but not CR between midostaurin arm and placebo arm [44]. Moreover, the survival benefits were not different based on the kind of FLT3 aberrations, including high FLT3-ITD, low FLT3-ITD, and FLT3-TKD (Table 1). The data allowed the approval of midostaurin for the management of AML patients with FLT3 mutations.

### 5.3. Lestauritinib

Lestauritinib is an orally administered polyaromatic indolocarbazole, which was grown as an inhibitor of tropomyosin receptor kinase. Lestauritinib could target TrkA, TrkB, and FLT3 and also inhibits JAK2 and JAK3 [23,46,47].

In a study, AML patients with FLT3 mutations were randomized to receive lestaurtinib or to receive a placebo during intensive chemotherapy. No difference was seen between lestauritinib and placebo in 5-year OS (46% vs. 45%; *p* = 0.3) and 5-year RFS (40% vs. 36%; *p* = 0.3) (Table 1) [48]. In a survey analysis, improved OS with concomitant azole antifungal use was associated with sustained FLT3 inhibition of >85%. The findings indicate clinical benefit according to the degree of FLT3 inhibition. Although impressive overall, the remission rates were similar between the cohorts (*p* = 0.4).

### 5.4. Crenolanib

Crenolanib is FLT3 inhibited against ITD and TKD mutations. Moreover, it could inhibit PDGFR but not c-KIT [49]. The dual FLT3 inhibitory characteristic is interesting because it was observed that relapse after an initial response to an FLT3 inhibitor could appear from acquired TKD mutations such as D835 and F691 positions. The results showed high efficacy in treating patients with newly diagnosed FLT3-mutated AML, showing an ORR of 96% (CR, 88%) and an OS rate of 88%, with a median follow-up of 6.2 months (Table 1) [50].

**Table 1 ijms-23-12708-t001:** Clinical trials of FLT3 inhibitors as front-line therapy in AML patients.

Agent (Reference Number)	Study Design and Population	Therapeutic Schedule	Treatment Outcomes	Adverse Effects
**First generation FLT3 inhibitors**				
**Sorafenib** [37]	Phase II (*n* = 276) ND AML, ≤60 yrs FLT3 mutation, not required Median age, 50 yrs	Induction -Cytarabine 100 mg/m^2^/d D1–7 -Daunorubicin 60 mg/m^2^/d D3–5 -Sorafenib 400 mg bid D10–19 vs. placebo Consolidation (or Allo-SCT) -HDAC +/− sorafenib 400 mg bid D8 to 3 days before next cycle Maintenance sorafenib 400 mg bid vs. placebo	In the overall group, in placebo vs. sorafenib, mOS, not different 3 yr-EFS 22% vs. 40% (*p* = 0.013) In FLT3-ITD+ group, mEFS (5 vs. 6 mo), mRFS (18 vs. 6 mo) and mOS (not reached vs. 19 mo)—all not different → ≤60 yrs, sorafenib with standard CTx → anti-leukemic effect.	Diarrhea Bleeding Cardiac event Hand-foot-skin reaction
**Sorafenib** [38]	Phase III (*n* = 211) ND AML Age > 60 yrs FLT3 mutation not required Median age, 68 yrs	Standard induction/consolidation +/− sorafenib Cytarabine, 100 mg/m^2^/d, D1–2 Daunorubicin 60 mg/m^2^/d. D3–5 Sorafenib 400 mg bid on D3 vs. placebo	CR, sorafenib vs. placebo → 48% vs. 60% (*p* = 0.12) ORR, 57% vs. 64% (*p* = 0.34) In FLT3-ITD+, CR 40% vs. 72% (NS) Overally, median EFS, 5 vs. 7 mo (*p* = 0.88) Median OS 13 vs. 15 mo (*p* = 0.12) → standard CTx with sorafenib, not beneficial in elderly AML pts.	Not determined
**Midostaurin** [43]	Phase IB (*n* = 69) Age of AML, 18–60 yrs, KPS ≥ 70 FLT3 mutation not required Median age, 39 yrs	Cytarabine 200 mg/m^2^/d, D1–7 Daunorubicin, 60 mg/m^2^/d, D1–3 Midostaurin 50/100 mg bid D1–7 → D15–21 or D8–21	Overally, CR—80% In FLT3 mutation+ → CR 12/13 (92%) In FLT3 mutation+ → 1-yr-DFS—50%/1-yr-OS—85% 2-yr-OS—62%	Nausea, vomiting, Diarrhea
**Midostaurin** [44]	Phase III (*n* = 717) Age of AML, 18–59 yrs FLT3 mutation, required Median age, 47.9 yrs	Cytarabine 200 mg/m^2^/d, D1–7 Daunorubicin 60 mg/m^2^/d D1–3 Midostaurin or placebo—D8–21 Midostaurin D 8–21 with HDAC Midostaurin maintenance for 12/28 for 28-day cycles	CR, 59% in midostaurin vs. 54% in placebo (*p* = 0.15) mOS, 74.7 mo vs. 25.6 mo (*p* = 0.009) 4-yr OS, 51.4% vs 44.3% (NS) mEFS, 8.2 vs. 3.0 mo (*p* = 0.002) 4-yr EFS, 28.2% vs. 20.6% (NS)	Anemia, rash, nausea
**Lestaurtinib** [48]	Phase III (*n* = 500) ND AML/high risk MDS Suitable for intensive Tx Presence of FLT3 mutation,	Various intensive Tx Lestaurtinib 80 mg bid starting, Increase to 100 mg bid If tolerable.	ORR—not different, lestauritinib group vs. control 5-yr OS, 46% in lestauritinib vs. 45% in control (*p* = 0.3) 5 yr-RFS, 40% vs. 36% (*p* = 0.3) Lestaurtinib with IC, feasible but no clinical benefit	Nasea, Diarrhea, Bilirubin ↑
**Second generation FLT3 inhibitors**				
**Crenolanib** [50]	Phase II (*n* = 26) Newly diagnosed AML, ≥18 yrs FLT3 mutations, required Median age, 55 yrs	Induction Cytarabine 100 mg/m^2^/d, D1–7 Daunorubicin 90 mg/m^2^/d or idarubicin 12 mg/m^2^/d D1–3 crenolanib 100 mg tid start on D9 until 72 hrs before next cycle	ORR, 96%, (CR, 88%) During median F/U 6.2 mo, OS rate, 88%	Periorbital edema, LFT elevation, Nausea, Rash

ND, newly, diagnosed; R/R, relapsed/refractory; CR, complete remission; CRp, Complete Remission with incomplete platelet recovery; DFS, disease-free survival; OS, overall survival; ORR, overall response rate; HDAC, high-dose ara-C; RFS, relapse-free survival.

## 6. FLT3 Inhibitors in Patients with Relapsed/Refractory Disease Unsuitable for Standard Therapy

### 6.1. Sorafenib

Ravandi et al. showed the efficacies of combined therapy with sorafenib and 5-AZA in 43 AML patients. In the data, 40 patients had FLT3-ITD mutations [51]. In all patients, 5-AZA was intravenously administered at 75 mg/m^2^/day for seven days and oral sorafenib was continuously administered. The response rate (RR) was 46%, the CR rate with incomplete count recovery (CRi) was 16%, and the partial response (PR) rate was 3% (Table 2). The combination therapy was well-tolerated, although several patients experienced grade 3–4 rashes.

Another study also showed the efficacy and tolerability of combination therapy in newly diagnosed AML patients with FLT3 mutations inappropriate to standard chemotherapy [52].

FLT3-mutated AML patients who received frontline therapy were analyzed. Twenty-seven patients with untreated AML with FLT3 mutations (median, 74 years) were enrolled in 2 separate protocols of 5-AZA with sorafenib. The ORR was 78% (CR, *n* = 7 [26%]; CRi (complete response with incomplete count recovery)/CRp (CR except for incomplete platelet recovery), *n* = 12 [44%]; PR, *n* = 2 [7%]). The patients were treated during a median of 3 therapeutic cycles (1– 35 cycles). The median remission duration of CR/CRp/CRi was 14.5 months (range, 1.1–28.7 months). Three patients including 1 with CR and 2 with CRi, received allo-SCT. The median OS for the entire group was 8.3 months, and 9.2 months in 19 responders. The regimen was well-tolerated by elderly patients with untreated FLT3-mutated AML with no early deaths.

### 6.2. Midostaurin

In a phase II study, the patients received midostaurin (50 mg bid or 100 mg bid) in patients with R/R AML or high-risk MDS with FLT3 mutations or WT [53]. In the data, none of the patients achieved CR. However, the blast counts were reduced in 71% of the FLT3-mutated AML patients and 42% of those without the mutations. The data showed the clinical impact of midostaurin in both patients with FLT3 mutations and WT (Table 2).

In the other phase I/II study, the combination of midostaurin and the hypomethylating agent AZA in 54 untreated and R/R AML and high-risk MDS patients showed a modest overall response rate of 26% (1/54 CR, 6/54 CRi, 6/54 morphologic leukemia-free state, and 1/54 PR) [41]. The median response duration was 20 weeks, and the median overall survival was 22 weeks at a median follow-up of 15 weeks (range, 1–85 weeks). The longest response duration was noted in patients without prior exposure to FLT3 inhibitors and patients who did not have a previous SCT.

### 6.3. Gilteritinib

Gilteritinib is a dual FLT3/AXL inhibitor with clinical effects against TKD mutations, but without working against KIT. The increased Axl-1expression is also associated with a resistance mechanism to other FLT3 inhibitors, such as midostaurin and quizartinib [42].

In comparative phase I/II data in dose-escalation and dose-expansion cohorts with R/R AML patients received gilteritinib, it was administered once a day in dose-escalation and dose-expansion cohorts (20 mg, 40 mg, 80 mg, 120 mg, 200 mg, 300 mg, or 450 mg) [54]. In the expansion cohort, 120 mg and 200 mg gliteritinib were administered to R/R AML patients with FLT3 mutations. Twenty-three patients were present in the dose-escalation cohort and 229 patients were in the dose-expansion cohort. In the data, ORR was 40% (Table 2). Dose-limiting adverse effects of grade 3 diarrhea and elevated aspartate aminotransferase (ALT) levels at 450 mg/day were observed.

To investigate the clinical efficacy of gilteritinib in a phase I study in Japanese patients with R/R AML, gilteritinib was administered in daily escalating doses in 6 cohorts (20, 40, 80, 120, 200, and 300 mg/day) [55]. In one patient, grade 3 tumor lysis syndrome was seen at 120 mg/day. In two patients, grade 3 elevated lactate dehydrogenase, amylase, and creatine phosphokinase levels, and syncope occurred at 300 mg/day. Of the five FLT3-mutated patients, ORR was 80%. In this study, 120 mg/day of gilteritinib was established as the recommended dose in Japanese patients.

The phase III ADMIRAL trial was compared with gilteritinib and salvage chemotherapy in 371 R/R AML patients with FLT3 mutations [56]. The median age of the 371 patients was 62 years (range, 19–85 years). Gilteritinib (*n* = 247) was given at 120 mg/day, and the patients were randomized 2:1 to one of four salvage chemotherapy regimens (LDAC, AZA, MEC, or FLAG-IDA) (*n* = 124). In the interim analysis of the gilteritinib group, the CR/CRh (CR/complete remission with partial recovery of peripheral blood counts) rate was 21%, the CR rate was 11.6%, and the CRh rate was 9.4% (Table 2). The median DOR was 4.6 months (range, 0.1–15.8 months). In the final analysis, the gilteritinib group hadsignificantly higher OS than the SC group (9.3 months vs 5.6 months, *p* = 0.0007). Additionally, the CR/CRh rate in the gilteritinib group was higher than in the SC group (34% vs. 15.3%, *p* = 0.0001).

### 6.4. Quizartinib

Quizartinib is a second-generation TKI that has more selected activity for FLT3 than first-generation inhibitors. However, it has still non-specific efficacy to other targets such as c-KIT and PDGFR. The excellent efficacy of quizartinib as an FLT3 inhibitor was proved in clinical data of R/R AML groups that received single-agent quizartinib.

Quizartinib was shown to have a highly selective impact on FLT3 in both FLT3-ITD-mutated and WT cell lines [57,58,59]. A phase I dose-escalation trial of quizartinib was investigated in 76 R/R AML patients, regardless of FLT3-ITD mutation status (Table 2) [60]. In the data, ORR in with FLT3-ITD mutated group was higher than WT group (53% vs. 14%, respectively). The data demonstrated that quizartinib has more specific activity in patients with R/R AML with FLT3-ITD mutations [61].

In addition, a randomized, open-label, phase IIb study evaluated to perform the potency and security of quizartinib in 76 R/R AML patients with FLT3-ITD mutations who received second-line remedial therapy or transplantation [62]. The patients were randomized to take quizartinib at 30 mg/day or 60 mg/day. In the data, both groups had CRc (composite CR, CR + CRp + CRi) rate of 47%. The ORR of each group was 61% and 71% and the continuation of CRc was 4.2 and 9.1 weeks, and the median OS time was 20.7 and 25.4 weeks, respectively (Table 2). The results showed that 60 mg/day was more effective than 30 mg/day.

In addition, a phase II trial investigated the combined therapy of quizartinib with low-dose cytarabine (LDAC) or 5-AZA in numerous untreated and R/R myeloid neoplasms including AML, myelodysplastic syndrome, and chronic myelomonocytic leukemia. In the interim analysis, an ORR was 67% (77% in 5-AZA, 23% in LDAC) and the median survival was 14.8 months (not reached in 5-AZA, 7.5 months in LDAC). Based on these results, it was concluded that quizartinib had efficacy in patients with FLT3-ITD-mutated myeloid neoplasms without D835 mutations [63]. However, a phase III study is needed to confirm these findings. Numerous phase I or II combination studies in newly diagnosed AML patients with FLT3-ITD mutations with quizartinib are summarized in Table 2.

### 6.5. Crenolanib

Crenolanib is a second-generation FLT3 inhibitor capable of inhibiting both FLT3-ITD and TKD mutations, with other signaling activity against c-KIT and PDGFR [63,64,65]. In a phase I trial in R/R FLT3 mutated AML, patients received idarubicin 12 mg/m^2^ for 3 days with HDAC, 5 g/m^2^ for 4 days, or 3 days in older than 60 years), followed by crenolanib starting on D5 and continued until 72 h [66]. The ORR of 36% was achieved in 18 patients without previous exposure to FLT3 inhibitors. The ORR was 36% in 36 patients who had previously received FLT3 inhibitors.

In a phase II clinical trial, crenolanib was administered at 200 mg/m^2^/d 3 times in 10 patients with R/R AML who progressed after SCT [63]. After receiving the drug, the ORR was 47%. Interestingly, crenolanib was shown to have synergistic anti-leukemic activity.

Iyer et al. reported the results of eight patients with R/R AML who received combination therapy with crenolanib and high-dose cytarabine/mitoxantrone. Four patients achieved CR/CRi after one cycle (CR, 2; CRi, 2). Only one patient showed a transient elevation in total bilirubin levels (Table 2) [67].

Maro et al. used salvage idarubicin and HDAC and crenolanib to treat patients with R/R FLT3-mutated AML. The ORR was 36% and the median OS was 259 days [68]. No dose-limiting toxicities were observed. Grade I gastrointestinal toxicities including nausea, vomiting, diarrhea, and abdominal pain were the major non-hematological AEs.

**Table 2 ijms-23-12708-t002:** Clinical trials of FLT3 inhibitors in relapsed or refractory disease or patients not suitable for standard therapy.

Agent (Reference Number)	Study Design and Population	Therapeutic Schedule	Treatment Outcome	Adverse Effects
**First generation FLT3 inhibitors**				
**Sorafenib** [51]	Phase I/II (*n* = 43) FLT3 mutation, not required Median age, 64 yrs	AZA 75 mg/m^2^/d D1–7 Sorafenib 400 mg bid	ORR 46%, CR 16%, CRi 27%, PR 3% DoR of CR/CRi, 2.3 mos Median OS—6.2 mos → Sorafenib with AZA, effective in relapsed AML pts with FLT3-ITD (+)	Fatigue, LFT elevation, Diarrhea
**Sorafenib** [52]	Phase II (*n* = 27) FLT3 mutated Unsuitable for standard CTx Median age, 74 yrs (61–86 yrs)	AZA 75 mg/m^2^/d D1–7 Sorafenib 400 mg bid	ORR, 78% (CR, 26%; CRi/CRp, 44%; PR, 7%) mDoR—14.5 mos 3 pts, received allo-SCT OS—8.3 mo in entire group, 9.2 mos in responder → the regimen, well tolerable in elderly pts with FLT3 (+)	Infection, Hyperbilirubin Anemia, Diarrhea, Fatigue
**Midostaurin** [53]	Phase IIB (*n* = 95) AML, MDS (RAEB, CMML FLT3 mutations, not required 64%, ≥ 65 yrs	Midostuarin, 50 or 100 mg bid	FLT3 mutation +—blast reduction, 71% FLT3-WT—blast reduction, 49% CR/Cri—0; PR—1/35 Blast reduction (≥50%)—49% → midostaurin, clinical efficacy in both pts with FLT3 (+) and WT.	Nausea, vomiting
**Midostaurin** [41]	Phase I/II (*n* = 54) AML, high risk MDS FLT3 mutation, not required Median age, 65 yrs	AZA D1–7 and Midostaurin 25 mg bid (cohort I) or 50 mg bid (cohort II) (MTD 50 mg bid)	ORR—26% (CR 1/54, CRi 6/54, MLFS, 6/54, PR 1/54) mDoR—20 wks -pts not exposed FLT3 inhibitor, longer (*p* = 0.05) -pts not received SCT, longer (*p* = 0.01) mOS—22 wks → midostaurin with AZA, effective and safe in AML and high-risk MDS pts	Neutropenia, thromvocytopenia Anemia, EF reduction Diarrhea Nausea/vomiting
**Second generation FLT3 inhibitor**				
**Gilteritinib** [54]	Phase I/II (*n* = 265), R/R setting FLT mutation, not required Median age, 64 yrs	Dose-escalation cohort vs. Dose-expansion cohorts (120–200 mg, given) -MTD of gilteritinib, 300 mg/d	ORR—40% → Gilteritinib—well tolerable	Diarrhea, Anemia, Fatigue, LFT ↑
**Gilteritinib** [55]	Phase I (*n* = 24) Japanes patient with R/R AML	Dose-escalating, 20/40/80/120/200/300 mg MTD of gilteritinib, 200 mg	ORR in FLT3 (+)—80% ORR in FLT3 WT—4/11 (36.4%) -120 mg/d gilteritinib, recommend → gilteritinib, well tolerated and effective in Japanese R/R AML pts.	Grade 3 LDH ↑, Amylase ↑ Syncope
**Giltertinib** [56]	Phase III (*n* = 371) R/R AML with FLT3 mutation	2:1 ratio received gliteritinib, 120 mg/d or 1/4 salvages—MEC, FLAG-IDA, LDAC, AZA	OS in gilteritinib ↑—SC group (9.3 vs. 5.6 mos; *p* < 0.001) CR/CRh, gilteritinib > SC group (34 vs. 15.3%, *p* = 0.001) → Gilteritinib, longer survival and higher remission rate than salvage CTx in R/R pts.	Cytopenia, QTc prolongation Pancreatitis, PRES, Differentiation syndrome
**Quizartinib** [60]	Phase I (*n* = 76) patients with R/R AML FLT3 mutation, not required Median age, 59.5 yrs	MTD of quizartinib 200 mg/d	In 17 FLT3-ITD (+) pts, -2CR, 3CRp, 5CRi, 13PR → 23 pts. In 37 FLT3-ITD (-) pts -2CRp, 3PR → 5 pts. In 22 FLT3 intermediate/not-tested status -1CR, 1CRp, 5PR → 7 pts.	Nausea, Vomiting, QTc prolongation,
**Quizartinib** [62]	Phase II (*n* = 76) R/R FLT3-ITD mutated AML after second-line or allo-SCT	30 mg/d (A) or 60 mg/d (B) Phase II (*n* = 76) R/R FLT3-ITD (+) AML after second-line or allo-SCT	CRc, 47%, DoR—22–26 wks 30 mg/d group—ORR, 61%; mOS—20.7 wks 60 mg/d group—ORR, 71%; mOS—25.4 wks	QTc prolongation, Nausea, Diarhea Vomiting
**Quizartinib** [63]	Phase II (*n* = 52) AML, high-risk MDS, CMML, FLT3-ITD required for enrollment Median age, 67 yrs	AZA, 75 mg/m^2^ SC/IV for 7 days LDAC, 20 mg SC twice daily 10 days Quizartinib, 60 or 90 mg	Response, 35 → 8 of LDAC arm (23%), 27 of AZA arm (77%) ORR, 67% (CR-8, CRp-7, CRi-18, PR-18, PR-2) ORR, 73%—FLT3-ITD+ (*n* = 48) 11 received to allo-SCT	Hypokalemia, Hypotension, Hypophosphatemia, Hyponatremia, QTc prolongation
**Crenolanib** [66]	Phase I (*n* = 13) R/R FLT3 mutated AML Median age, 51 yrs	Idarubicin 12 mg/m^2^/d D1–3 Cytarabine 1.5 g/m^2^/d D1–4 Crenolanib 60–100 mg tid start on D5 → continued until 72 before next cycles	ORR—36% (CR, 1/CRi, 3) mOS—259 days → full-dose crenolanib, safely combined with idarubicin and HDAC in R/R AML pts.	Nausea, Vomiting, Diarrhea, Abdominal pain
**Crenolanib** [67]	Phase II (*n* = 8), R/R FLT3 mutation, not rquired Median age, 64 yrs	HAM—Cytarabine 1.0 g/m^2^/d D1–6 and mitoxantrone 10 mg/m^2^ D1–3 vs. Crenolanib 100 mg tid start on D8	CR—2/6, CRi—2/6 → full-dose crenolanib, well-tolerable with HAM in R/R elderly AML pts.	AST/ALT elevation

AZA, azacitidine; ORR, overall response rate; CRi, complete remission with incomplete count recovery; PR, partial remission l DOR, duration of remission; OS, overall survival; CRp, Complete Remission with incomplete platelet recovery; MLFS, median leukemia-free survival; DOR, duration of response; EF, ejection fraction; MEC, mitoxantrone, etoposide, cytoxane; HDAC, high-dose ara-C; FLAG, fludarbine, ara-C; all-SCT, allogeneic stem cell transplantation; HI, hematologic improvement; EFS, event-free survival; NR, no response.

## 7. Maintenance Therapy for FLT3-ITD-Mutated AML

### 7.1. Midostaurin

In the recent phase II randomized RADIUS trial, maintenance therapy of midostaurin following allo-SCT did not appear to improve the prognosis of the SOC therapy with or without midostaurin after allo-SCT for AML patients with FLT3 mutations (Table 3) [69]. In the data, 60 adult patients (aged 18–70 years old) patients who received allo-SCT in first CR, were randomly assigned to receive SOC with madostaurin or SOC only (50 mg twice daily). The trial did not reach its primary endpoint because RFS was similar between the two groups (89% in the Midostaurin group vs. 76% in SOC group; HR, 0.46 [95% CI, 0.12–1.86]; *p* = 0.27). The rate of adverse events including GVHD was also similar between the two groups.

Moreover, a recent ongoing phase II hypothesis-generating trial is being investigated to determine whether adding midostaurin to intensive chemotherapy followed by allo-SCT and single-agent maintenance therapy for 12 months was feasible and favorably influenced EFS compared to the control group. Patients 18 to 70 years of age with newly diagnosed AML and centrally confirmed FLT3-ITD were eligible for the study (NCT01477606).

Two hundred and eighty-four patients were treated, including 198 younger (18–60 years) and 86 older (61–70 years) patients. The CR/CRi rate was 76.4% after induction therapy. The majority of the patients (72.4%) in CR/CRi received allo-SCT. Maintenance management was initiated in 97 patients (34%), including 75 after allo-SCT and 22 after consolidation with HDAC. The median time of maintenance therapy was 9 months after allo-SCT and 10.5 months after HDAC. The 2-year EFS and OS rates were 39% and 34% in younger patients, and 53% and 46% in older patients, respectively.

### 7.2. Sorafenib

Whether maintenance therapy using FLT3 inhibitors, such as the multitargeted tyrosine kinase inhibitor sorafenib, increases prognosis after allo-SCT is not well-known. In a randomized, placebo-controlled, double-blind phase II trial, 83 adult patients with FLT3-ITD mutated AML in CR after SCT were randomized to receive either the multitargeted and FLT3-kinase inhibitor sorafenib (*n* = 43) or placebo (*n* = 40) for 24 months [70]. With a median follow-up of 41.8 months, the 24-month RFS probability was 53.3% (95% CI: 0.36 to 0.68) in the placebo group vs 85.0% (95% CI: 0.70 to 0.93) in the sorafenib group (HR = 0.256; 95% CI: 0.10 to 0.65; *p* = 0.002). The data showed that patients with undetectable minimal residual disease (MRD) before SCT and those with detectable MRD after SCT has the strongest benefit from sorafenib. Maintenance therapy of sorafenib reduced the risk of relapse and death after SCT for FLT3-ITD mutated AML (Table 3).

Xian et. performed a randomized phase III trial at seven hospitals in China. Eligible patients (range, age 18–60 years) had AML with FLT3-ITD mutations, received allo-SCT, had an ECOG performance status below grade 2, were in a CRc state before and after SCT, and had hematopoietic recovery within 60 days of SCT (Table 3) [71]. The patients were randomized to receive sorafenib maintenance (400 mg bid/day) or were in a control group.

Two hundred and two patients were randomized to be assigned to sorafenib group (*n* = 100) or control groups (*n* = 102). The 1-year cumulative incidence of relapse was 7.0% (95% CI: 3.1–13.1) in the sorafenib group and 24.5% (95% CI: 16.6–33.2) in the control group (HR = 0.25, 95% CI: 0.11 to 0.57; *p* = 0.0010). Within post-SCT duration, the most common grade 3 and 4 adverse effects were infections (25 of 100 patients in the sorafenib group vs. 24 of 102 patients in the control group, acute graft-versus-host disease (GVHD) and 23 of 100 patients vs. 21 of 102 patients), and chronic GVHD (18 of 99 patients vs. 17 of 99 patients). Post-SCT maintenance therapy of sorafenib after SCT reduced relapse and was well-tolerated in AML patients with FLT3-ITD mutations.

## 8. Resistance Mechanisms to FLT3 Inhibitor

Mechanisms of primary and secondary resistance to FLT3 inhibitors in AML cells with FLT3 mutation are summarized in Figure 1.

The mechanism of primary intrinsic resistance to FLT3 inhibitors is the lack of dependency of AML with FLT3 mutations to FLT3 signaling due to the coexistence of numerous leukemic clones and low allelic cells of the FLT3 mutation [72]. The second mechanism of primary intrinsic resistance is the presence of mutations preventing interaction with specific agents, notably TKD mutations conferring resistance to type II FLT3 inhibitors [73].

Additionally, FLT3-ITD 627E mutation has been found that confers primary resistance to FLT3 inhibitors by upregulating the antiapoptotic protein Mcl-1 [74], and upregulation of the antiapoptotic protein, Bcl-xL, has been a demonstrated mechanism of resistance to FLT3 inhibitors in leukemic cells with FLT3 mutations [75].

Induction of new TKD mutations occurs as a common secondary intrinsic mechanism of resistance to type II FLT3 inhibitors, for example, in relapsed AML patients with FLT3-ITD mutations received quizartinib or sorafenib therapy [73,76]. Moreover, genomic instability is a frequent phenomenon in AML patients with FLT3-ITD mutations, treated with FLT3 inhibitors at relapse [77].

The oncogenic serine/threonine kinase Pim-1 is upregulated downstream of FLT3-ITD and potentiates FLT3 signaling in a positive feedback loop [44,46]. For example, in AML patients with FLT3-ITD mutations with acquired resistance to sorafenib, Pim-1, and the Pim kinase were found to be upregulated [78]. Inhibition of Pim kinases restored the sensitivity of FLT3-ITD cells to FLT3 inhibitors in a mouse model [79].

Increased fibroblast growth factor 2 (FGF2) secretion by the BM microenvironment after chemotherapy or FLT3 inhibitor therapy stimulates leukemic cells with FLT3 mutations via binding to fibroblast growth factor receptor (FGFR) 1 on leukemic cells [80]. Furthermore, BM stroma–mediated resistance results from enhanced CXCL12–CXCR4 mediated mechanism [81], at least in part due to Pim-1 overexpression, as Pim-1 phosphorylates CXCR4, enabling its cell surface translocation and expression [82].

## 9. Overcoming Strategy for Resistance of FLT3 Mutations

Venetoclax, an inhibitor of the anti-apoptotic protein Bcl-2, particularly showed interesting clinical data about combining therapy with FLT3 inhibitors. Preclinical data demonstrated that leukemic cells with FLT3-ITD mutations had higher expressed and upregulated Bcl-2 proteins compared to cells with FLT3-WT [83]. The proteasome inhibitor bortezomib was shown to promote the autophagosomal degradation of FLT3-ITD and was cytotoxic to FLT3-ITD positive cells, such as leukemic cells with FLT3-ITD+ resistant to quizartinib. Combination therapy with bortezomib, midostaurin, and chemotherapy had effective cytotoxic efficacy, but it has clinical toxicity [84]. The STAT5 inhibitor pimozide reduced the survival of FLT3-ITD-positive leukemic cells [85]. **Pimozide decreased tyrosine phosphorylation of STAT5 and induced apoptosis in leukemic cells. Moreover, it targets deubiquitinating DUB enzyme in leukemic cells indicating that the effects of pimozide on STAT5 activity might be indirect. Thus, this STAT5 inhibitor could inhibit growth of leukemic cells**. Additionally, the dual JAK/FLT3 inhibitor pacritinib demonstrated efficacy in a phase I study [86]. The mTOR inhibitors including everolimus could target and inhibit PI3K/AKT/mTOR pathway. The simultaneous inhibiting of both FLT3-ITD and mTOR could be effective in tumor suppression in upregulated acute leukemic cells resistant to single-agent FLT3 inhibitors [87]. Moreover, metformin could inhibit the PI3K/Akt/mTOR pathways and be shown to be active synergistically with sorafenib in leukemic cell lines with FLT3 mutations [88]. Ponatinib is permitted to target BCR-ABL in chronic myeloid leukemia but it is also a type II FLT3 inhibitor with clinical efficacy against F691L. In a phase I study, ponatinib in the first application showed modest efficacy in patients treated heavily with R/R AML [89]. Cabozantinib is a multi-kinase inhibitor and has a selectively cytotoxic efficacy to leukemic cells with FLT3-ITD mutations. A phase I trial showed that the drug sustained the inhibition of leukemic cells with FLT3-ITD and F691 mutations [90]. Another multi-kinase inhibitor, pexidartinib, has been studied in various solid tumors and showed clinical efficacy against FLT3. Recently, a phase I/II study of pexidartinib in patients with R/R AML with FLT3-ITD mutations showed an ORR of 21% and a CR of 11% [91]. FF-10101 as a novel FLT3 inhibitor is beginning to be recognized as the first covalently binding FLT3 inhibitor. In preclinical studies, FF-10101 demonstrated potent activity against quizartinib-resistant AL and gatekeeper F691L mutations [92].

## 10. Conclusions

FLT3 inhibitors have become an essential component of the treatment for patients with FLT3-mutated AML. However, owing to rapid changes in practice, many unresolved issues exist.

First, there is insufficient data to determine which of several approved and unapproved FLT3 inhibitors is preferable. For example, midostaurin is used in combination with intensive chemotherapy for newly diagnosed patients as a *de facto* standard. However, second-generation FLT3 inhibitors may be more useful considering their property of more potent and selective FLT3 inhibition.

Second, in elderly patients unsuitable for intensive chemotherapy, 5-AZA with venetoclax has recently become the treatment of choice for a newly diagnosed subset of patients [93]. In this context, FLT3 inhibitors could have a feasible clinical impact on the therapeutic schedule. Thus, low-intensity combination therapy with the FLT3 inhibitors venetoclax and 5-AZA could have beneficial clinical impacts in ineligible patients with intensive chemotherapy.

Third, non-transplant patients received FLT3 inhibitors, while the transplant group received FLT3 inhibitors before or after allo-SCT. When comparing the two study groups, the prognosis of allo-SCT might be also improved by the clinical applications of FLT3 inhibitors. Therefore, novel FLT3 inhibitors to overcome common resistances are anticipated. FF-10101 is an FLT3 inhibitor covalently binding to the C695 residues of FLT3. It is not affected by F691L gatekeeper mutations and has the characteristic potential impact in quizartinib-resistant AML cells with F691 mutations [94]. Several highly selective FLT3 inhibitors with the potential to overcome resistances are in development.

Comprehensively, we tried to show the clinical data of FLT3 inhibitors for AML patients with FLT3 mutations, which were arranged and compared according to the frontline, the R/R setting, and maintenance therapy after allo-SCT in patients with FLT3-mutated AML.

## Figures and Tables

**Figure 1 ijms-23-12708-f001:**
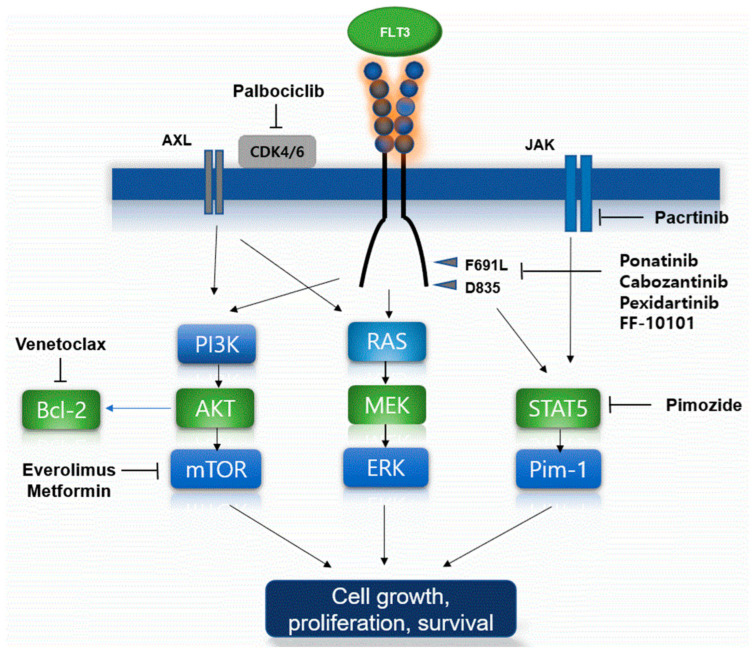
Proposed mechanisms of FLT3 inhibitor resistance and the established overcoming agents. Schematic of FLT3 inhibitor resistance mechanism including several types of FLT3 mutations were present. Several targeted agents such as venetoclax, proteasome inhibitors, multikinase inhibitors, novel dual agents, and novel FLT3 inhibits biologic agents, could overcome and thus lead to cancer cell growth, proliferation, and prolonged survival of the FLT3 mutations by several mechanisms.

**Table 3 ijms-23-12708-t003:** Clinical trials of FLT3 inhibitors as maintenance therapy after stem cell transplantation in AML patients.

Agent (Reference Number)	Study Design and Population	Therapeutic Schedule	Treatment Outcomes	Adverse Effects
**First generation FLT3 inhibitors**				
**Midostaurin** [69]	Phase II (*n* = 60), 18–70 yrs FLT3 mutation, required	Midostaurin -50 mg bid/d for 12 d in 4-wk cycle	18-mo RFS—89% in midostaurin arm vs, 76% in Standard-Of-Care arm (*p* = 0.27). some pts with higher levels of FLT3 inhibition -prolonged RFS (*p* = 0.06) and improved survival (*p*= 0.048) → midostaurin, clinical benefit in some FLT-ITD+ pts.	Vomiting, Nausea, Fatigue, Diarrhea
**Sorafenib** [70]	Phase II (*n* = 83) FLT3-ITD+ pts in CR after SCT	Sorafenib, 200–800 mg/d, +60–+100 d after SCT. during 24 months, Tx—continuously at 24 mo	Relapse/death risk, lower in sorafenib arm vs. placebo arm (HR = 0.39, *p* = 0.013) 24 mo-RFS 53.3% vs. 85.0% (HR = 0.256; *p* = 0.002) → Sorafenib maintenance, reduce risk of relapse and death after SCT for FLT-ITD+ AML.	
**Sorafenib** [71]	Phase III (*n* = 202), 18–60 yrs	400 mg bid/d at 30–60 post-SCT.	1-yr cumulative relapse in sorafenib arm, 7.0% vs. 24.5% in control arm (*p* = 0.0010) → sorafenib maintenance, reduce relapse and well-tolerable	Infection, Acute/chronic GVHD, Hematologic toxicity
**Midostaurin** (NCT01477606)	Phase II (*n* = 284), 18–70 yrs 18–60 yrs (*n* = 198) 61–70 yrs (*n* = 86)		CR/Cri—76.4% (younger, 75.8%/older, 77.9%) 2-yr CIR in SCT (18.1% and 17.6% in younger and older) -lower than CTx alone (39.2% and 56.4%) 2-yr CIR in maintenance group, 13.3% -lower than HDAC CTx alone 43.5% (*p* = 0.02)	QTc prolongation Lung toxicity, Diarrhea, Mucositis, Cytoepnia

RFS, relapse-free survival; HR, hazard ratio; SCT; relapse-free survival; SCT, stem cell transplantation; HR, hazard ratio; CRi, complete remission with incomplete count recovery; GVHD, graft-versus-host disease; CIR, the cumulative incidence of remission; HDAC, high dose ara-C.

## Data Availability

Not applicable.

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
