# Peer review of "Clinical Efficacies of FLT3 Inhibitors in Patients with Acute Myeloid Leukemia"

_ijms, 2022, doi:10.3390/ijms232012708_

Round 1
Reviewer 1 Report
Manuscript “Clinical considerations for use of FLT3 inhibitors in patients
with acute myeloid leukemia: applications of several circumstances” by Moo-Kon Song, Byeong-Bae Park, and Ji-Eun Uhm provide a comprehensive overview about currently know FLT3 inhibitors and mechanisms of resistance development against targeted therapies in AML.
Major points:
1. Chapter on resistance development need to be elaborated further including novel literature.
2. Pimozide is not a selective STAT5 inhibitor. Multiple studies using actual STAT5 inhibitors on AML we already conducted. They have to be addressed properly.
3. Page 2 – duplicated domains of FLT3 – incorrect wording.
4. Figure 1 – CD4/6 – probably CDK4/6 was meant.
5. Figure 1 not referenced in text.
6. Table 1 and 3 not referenced in text.
7. Page 1, 2. Acute leukemic cells – better term is leukemic blasts.
8. Stimulating of receptor – better term is activation of receptor.
9. Title and abstract of the review are confusing and do not correspond to the material presented. Please consider adapting.
10. Writing of the numbers in the text needs to be unified. Please do not start sentences with numbers, even if spelt out.
11. Logic in which drugs are presented and discussed is unclear, please elaborate on it.
12. Citation missing in the paragraph about sorafenib, vorinostat, bortezomib clinical study.
13. Careful proof reading of the whole manuscript required – many typos, not connected sentences, illogical connections.
Author Response
Manuscript “Clinical considerations for use of FLT3 inhibitors in patients
with acute myeloid leukemia: applications of several circumstances” byMoo-Kon Song,Byeong-Bae Park,andJi-Eun Uhmprovide a comprehensive overview about currently know FLT3 inhibitors and mechanisms of resistance development against targeted therapies in AML.
Major points:
1.Chapter on resistance development need to be elaborated further including novel literature.
(Answer) we made novel literature about resistance of FLT3
2.Pimozide is not a selective STAT5 inhibitor. Multiple studies using actual STAT5 inhibitors on AML we already conducted. They have to be addressed properly.
(Answer) we added some sentences about pimozide.
3.Page 2 – duplicated domains of FLT3 – incorrect wording.
(answer) we revised the word.
4.Figure 1 – CD4/6 – probably CDK4/6 was meant.
(answer) we revised the word
5.Figure 1 not referenced in text.
(answer) we inserted he word
6.Table 1 and 3 not referenced in text.
(answer) we inserted he word
7.Page 1, 2. Acute leukemic cells – better term is leukemic blasts.
(Answer) we changed the word.
8.Stimulating of receptor – better term is activation of receptor.
(Answer) we changed the word.
9.Title and abstract of the review are confusing and do not correspond to the material presented. Please consider adapting.
(Andwer) we changed some pf title and abstract.
10.Writing of the numbers in the text needs to be unified. Please do not start sentences with numbers, even if spelt out.
(Answer) some senstences were revised
11.Logic in which drugs are presented and discussed is unclear, please elaborate on it.
12.Citation missing in the paragraph about sorafenib, vorinostat, bortezomib clinical study. (answer) The sentence is omitted.
13.Careful proof reading of the whole manuscript required – many typos, not connected sentences, illogical connections.
(answer) we revised illogical typos.
Reviewer 2 Report
Authors described concise review of FLT3 inhibitor in AML.
1. Authors described FLT3 mutation is the most frequent mutation associated with poor prognosis. Whereas, FLT3-ITD mutation is associated with poor prognosis. Please correct it.
2. Authors mentioned FLT3 mutation allelic ratio. Whereas, the clinical impact became low. Please modify the description.
Author Response
- Authors described FLT3 mutation is the most frequent mutation associated with poor prognosis. Whereas, FLT3-ITD mutation is associated with poor prognosis. Please correct it.
(Answer) in some part, I revised the contents. Thank you.
- Authors mentioned FLT3 mutation allelic ratio. Whereas, the clinical impact became low. Please modify the description.
(Answer) AR data is relatively low in our manuscript. Thus, the clinical impact is low, please understand me. Thank you.
Reviewer 3 Report
ijms-1923862
Clinical considerations for use of FLT3 inhibitors in patients with acute myeloid leukemia : applications of several circum- stances.
The article " Clinical considerations for use of FLT3 inhibitors in patients with acute myeloid leukemia : applications of several circum- stances." (ijms-1923862) by Song MK, et al. reviewed that FLT3 inhibitors played an important role for treatment strategy of AML. This review article was informative and organized well. However, there are minor issues in this article.
1. Tables was very well organized, but it was not easy to read because the letters seemed to be relatively large. I considered that each table could be kept within one page if possible.
2. FLT3 inhibitors is a key drug for AML with FLT3 mutation. Therefore, several clinical trial of FLT3 inhibitors are ongoing. I recommend that the table about ongoing clinical trials about FLT3 inhibitors is added.
Author Response
1.Tables was very well organized, but it was not easy to read because the letters seemed to be relatively large. I considered that each table could be kept within one page if possible.
(answer) too many studies are present. Thus, I could not include the studies. please understand me. Thank you.
- FLT3 inhibitors is a key drug for AML with FLT3 mutation. Therefore, several clinical trial of FLT3 inhibitors are ongoing. I recommend that the table about ongoing clinical trials about FLT3 inhibitors is added.
(Answer) As you indicated, ongoing clinical trials is so important, However, so many data with result is present. Thus, I could not included the study. please, understand me. Thank you.
Round 2
Reviewer 1 Report
Thank you for revising the mauscript.